# Tuning the Catalytic Activity of a Quantum Nutcracker for Hydrogen Dissociation

**Lei Tao [1,2], Yu-Yang Zhang [1] , Sokrates T. Pantelides [1,2] and Shixuan Du [1,3,*]**

[1]   Institute of Physics & University of Chinese Academy of Sciences, Chinese Academy of Sciences, Beijing 100190, China; ltao@iphy.ac.cn (L.T.); zhangyuyang@ucas.ac.cn (Y.-Y.Z.); pantelides@Vanderbilt.Edu (S.T.P.)

[2]   Department of Physics and Astronomy and Department of Electrical Engineering and Computer Science, Vanderbilt University, Nashville, TN 37235, USA

[3]   Songshan Lake Materials Laboratory, Dongguan 523808, Guangdong, China

*   Correspondence: sxdu@iphy.ac.cn

**Abstract:** A quantum nutcracker, a recently proposed catalytic system for hydrogen dissociation, consists of two inert components: an organic molecule such as a transition metal phthalocyanine and an inert surface such as Cu or Au. The reaction takes place at the interface between the two components, which are weakly bonded by Van der Waals (VdW) forces. Here, we explore a method used to tune the reaction barrier in a quantum nutcracker system for hydrogen dissociation. By employing density-functional-theory calculations, we find that the $H_2$ entry barrier, which is the rate-limiting barrier, is reduced by replacing the phthalocyanine by porphyrin derivatives such as octaethylporphyrin (OEP) and tetraphenylporphyrin (TPP). The system remains active if a dissociated H atom is adsorbed on the transition metal ion. Metallic two-dimensional materials such as $NbS_2$ and $CoS_2$ are good candidates for the quantum nutcracker. The present design of a quantum nutcracker for hydrogen dissociation provides new opportunities with which to induce catalytic activity in VdW-bonded systems.

**Keywords:** heterogeneous catalysis; hydrogen dissociation; Van der Waals bonded system; density-functional-theory calculations

## 1. Introduction

It is well known in heterogeneous catalysis that reactions are mediated through electronic interactions between reactant molecules and catalyst surfaces [1]. Significant efforts have been made to promote catalytic properties, including optimizing the local electronic states of active sites and increasing the surface-to-volume ratio [2–4]. In recent years, two-dimensional (2D) materials such as graphene and transition metal dichalcogenides have also been widely investigated for electrocatalysis [5–7], photocatalysis [8–10], and conventional heterogeneous catalysis [11]. However, the intrinsic electronic properties of 2D materials provide both opportunities and challenges for catalytic applications, because the surfaces of 2D materials are not as catalytically active as those of conventional noble metals. There are various routes to tuning the electronic states in 2D materials, including introducing edges [12], dopants [13], functional groups [14], and metal clusters or single atoms [15,16]. The structural modification and functionalization of 2D materials for catalysis has been the subject of extensive investigations [17].

Hydrogen dissociation on metal surfaces has been widely studied for several decades because it is an important step in hydrogenation reactions such as the hydrogenation of alkanes and alkynes [18]. It is now known that the best catalysts are expensive metals such as Pt, Pd, and Ir, which has motivated

extensive research on alternative catalytic approaches based on inexpensive materials. Recently, a model catalytic system called the "quantum nutcracker" was introduced [19], providing an alternative paradigm for the catalytic dissociation of hydrogen molecules. A quantum nutcracker is composed of two inert components, namely, manganese–phthalocyanine (MnPc) and a Au(111) or Cu(111) surface. The reaction takes place at the interface between the two components, which are weakly bonded by VdW interactions. $H_2$ molecules adsorb on the substrate and diffuse under the molecule where they are "cracked", with H atoms exiting on the surface. The vibrational magnitude of organic molecule plays an important role in the overall barrier. Indirect experimental evidence is consistent with predictions [20,21].

In this paper we review briefly the concept of a quantum nutcracker and report new investigations aiming to tune its catalytic activity by varying the organic molecules, the inert surface, or both, including investigations using 2D materials as the inert surface. By employing density functional theory (DFT)-based calculations, we find that the overall barrier can be reduced by modifying the configuration of the organic molecule, including by changing the functional groups attached to the porphyrin ring. The $H_2$ entry barrier, which is the highest barrier in the dissociation process, is significantly reduced by replacing transition metal phthalocyanine molecules with transition metal porphyrin derivatives such as octaethylporphyrin (OEP) and tetraphenylporphyrin (TPP). We also find that the quantum nutcracker is still active when a dissociated H atom adsorbs on top of the transition metal ion. A projected density of states shows that the *d* orbital of the transition metal ion plays a major role in the dissociation process. Furthermore, we replace the inert metal surface with representative 2D materials. Metallic 2D materials such as $CoS_2$ and $NbS_2$ are also good candidates. Our results provide a new way to improve the catalytic efficiency in a quantum nutcracker system.

## 2. Methods

DFT calculations were performed using the Vienna Ab Initio Simulation Package (VASP) with the projected augmented wave (PAW) method [22,23]. Wave functions were expanded on a plane-wave basis set to a 400 eV energy cutoff. The exchange and correlation effects have been described by the Perdew-Burke-Ernzerhof with Van der Waals density function (VdW-DF) of optB86b version [24,25]. The Au surface was modeled by a (8 × 8) supercell which consisted of a 4 layer Au slab with a vacuum layer of 15 Å. All atoms except the bottom two Au/Cu layers were fully relaxed until the net force was smaller than 0.02 eV/Å. Single-layer graphene, $NbS_2$, and $CoS_2$ were modeled by a (8 × 8) supercell. The reaction pathways were investigated using the climbing image nudged elastic band (CI-NEB) method [26,27]. Five images were inserted into the initial and final states. The spring force between adjacent images was 5.0 eV/Å, and images were optimized until the forces on each atom were less than 0.02 eV/Å. Because of the large dimensions of the supercell, the Brillouin zone was sampled with a single Γ point.

## 3. Results and Disscusions

A quantum nutcracker consists of two inert components: an organic molecule such as transition metal phthalocyanine (TMPc), and an inert surface such as Au(111). In Figure 1a we show a schematic describing the various stages of a quantum nutcracker for hydrogen dissociation. The reaction process has three steps, as elaborated in our previous paper [19]: hydrogen molecules diffuse on the surface (initial state (IS)) and enter the interface between those two components (intermediate state (IMS)), followed by a break of the hydrogen bond at the interface (final state (FS)). Finally, the dissociated hydrogen atoms leave the interface. Among them, the entry of $H_2$ in the molecule–surface interface, the first step, is the bottleneck of the overall reaction. In the second step, the dissociation barrier depends on the electronic structure of the transition metal atom in the organic molecule. In the third step, the energy barriers for the diffusion of atomic hydrogen on the surface are small and can be overcome at room temperature. In the case of transition metal phthalocyanines on Au(111), out of all

the 3*d* transition metal atoms, Mn is the best candidate. Suitable alternative organic molecules are porphyrin and their derivatives, such OEP and TPP, as shown in Figure 1b.

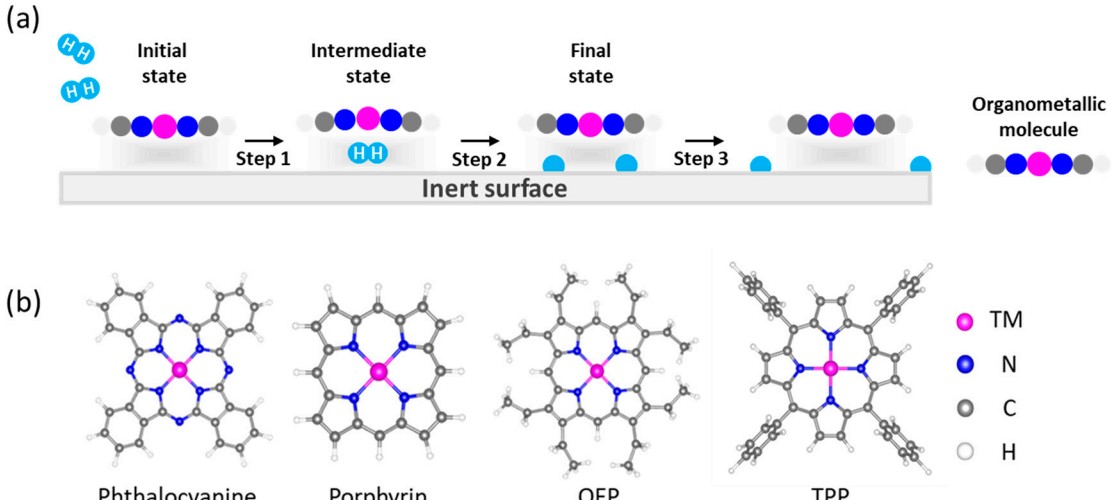

**Figure 1.** Schematics describing a quantum nutcracker for hydrogen dissociation. (**a**) A quantum nutcracker consists of two inert components as two jaws: an organic molecule and an inert surface. (**b**) The configurations of organic molecule candidates: transition metal phthalocyanine, transition metal porphyrin, transition metal octaethylporphyrin (OEP), and transition metal tetraphenylporphyrin (TPP).

Here, we use manganese as the representative transition metal and Au(111) as the inert substrate to investigate the hydrogen dissociation process of MnPP, MnOEP, and MnTPP on Au(111). When an $H_2$ molecule enters the interface it is trapped between the jaws of the quantum nutcracker (on top of an Au atom and underneath the Mn), which we define as the IMS. In Figure 2a we show the $H_2$ entry barrier, which is the highest barrier in the dissociation process, as a function of the Mn–Au distance in the IS. For Mn-porphyrin and its derivatives, the Mn–Au distance is larger than that in an MnPc/Au system. The $H_2$ entry barrier in the first transition state (TS1) significantly decreases as the Mn–Au distance increases. In Figure 2b,c, we show the $H_2$ dissociation barrier of Step 2 as a function of the Mn–Au distance in the transition states (TS2) and potential energy landscapes in the reaction, respectively. The $H_2$ dissociation reaction is exothermal in MnPc/Au and the Mn-porphyrin/Au system, and endothermic in MnOEP/Au and MnTPP/Au. The dissociation reaction barrier increases monotonically with the Mn–Au distance in TS2 but the dissociation barriers are still lower than the entry barriers in all Mn-organic molecule/Au(111) systems. Of note is that the largest entry barrier of MnPc on Au(111) (1.2 eV) is much smaller than the calculated binding energy (4.3 eV). Thus, in this process, molecules do not desorb from the surface. Hence, the overall barrier can be reduced by replacing the MnPc molecule with an Mn-porphyrin derivative such as OEP or TPP in the quantum nutcracker.

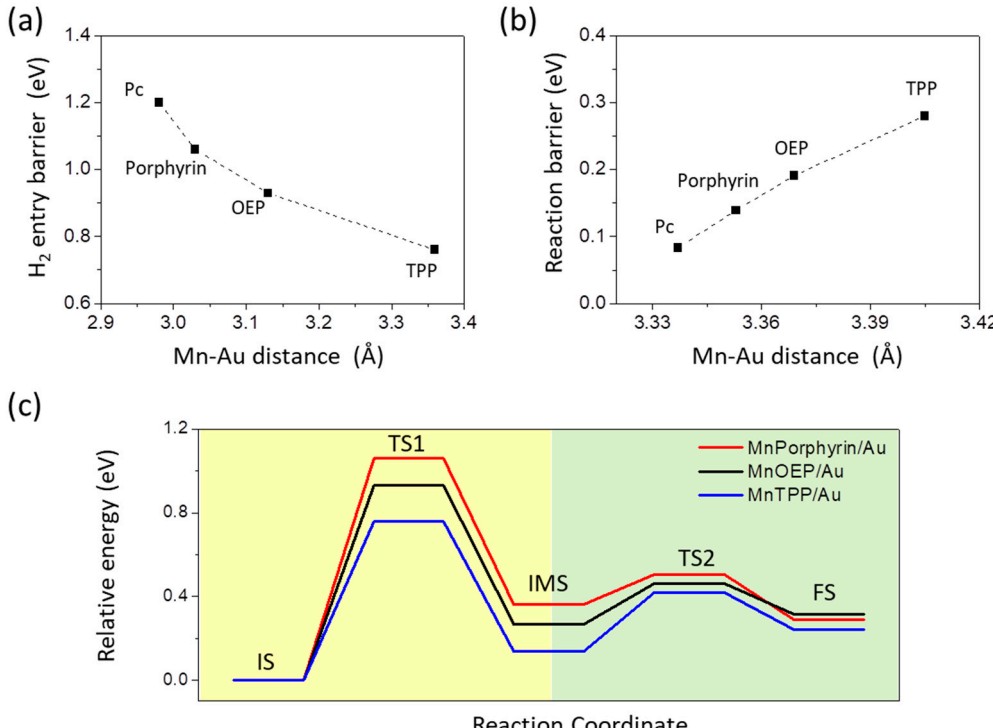

**Figure 2.** Relationship between catalytic activity and Mn–Au distance for the hydrogen dissociation reaction in an Mn-organic molecule/Au(111) system. (**a**) $H_2$ entry barrier as a function of Mn–Au distance in initial states. (**b**) $H_2$ dissociation barrier as a function of Mn–Au distance in Step 2. (**c**) Potential energy landscapes in different quantum nutcracker systems according to (**a**,**b**). Legend: IS, initial state; IMS, intermediate state; FS, final state; TS1, first transition state; TS2, second transition state.

Single atoms and small molecules, such as H resulting from the $H_2$ splitting process or contaminants such as CO or NO, may adsorb on top of TMPc molecules [21,28,29]. Indeed, the ligand attached to the transition metal would also influence the interaction between the organic molecule and the surface [20,30,31]. In Figure 3a we show the reaction process of different TMPcs with an adsorbed H atom on Au(111). In the H-MnPc/Au and H-CoPc systems, the reaction is a one-step process, while in H-FePc/Au, it is a two-step process. The TM-Au distances in the IS are 3.92 Å (H-MnPc/Au), 3.87 Å (H-FePc/Au), and 3.96 Å (H-CoPc/Au), respectively, indicating that the dissociation barrier also increases monotonically as the TM-Au distance increases. The projected densities of states (PDOSs) in the TS1 of the H-MnPc/Au systems on the hydrogen $s$ orbital, the Mn $d_{xz}$ orbital, and the Au $d_{xz}$ are shown in Figure 3b, and the partial charge density from −1.5 eV to the Fermi level is shown in Figure 3c. The orbital overlap between the $s$ orbital of the H atoms and the Mn $d_{xz}$ orbital shows that H–H bond breaking happens via the interaction of the hydrogen anti-bonding state with the transition metal ion.

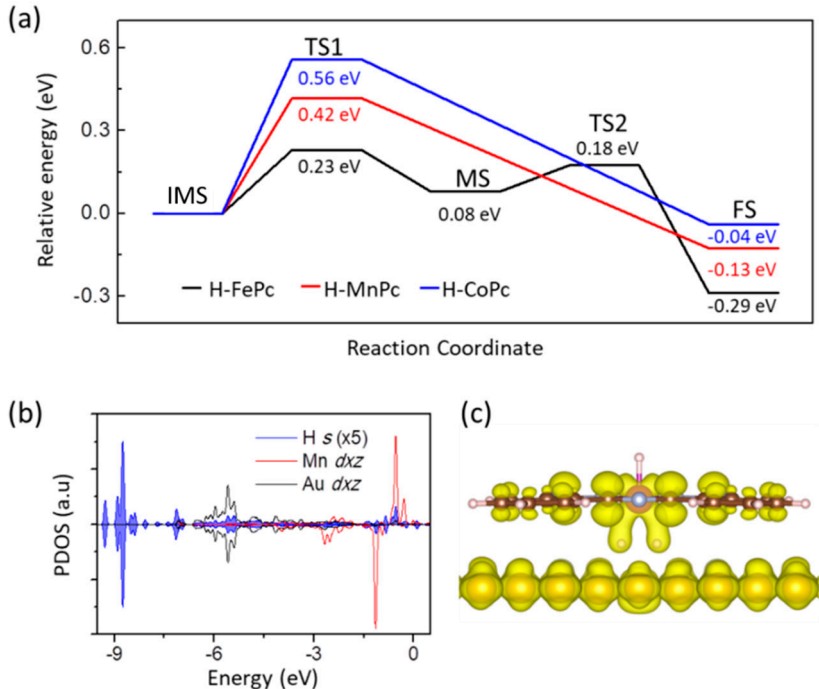

**Figure 3.** (**a**) Reaction pathway of hydrogen dissociation in H-TMPc/Au(111). (**b**) Projected density of states on hydrogen *s*, Mn *dxz*, and Au *dxz* in transition state. (**c**) Partial charge density from −1.5 eV to Fermi level shown in (**b**). Legend: PDOS, projected density of state.

Finally, we replaced the Au(111) surface with a monolayer of a two-dimensional material. We tried different candidates, namely, graphene and transition metal dichalcogenides such as $NbS_2$ and $CoS_2$, which are metallic. Reaction pathways are shown in Figure 4a and the corresponding side views of the intermediate and final states in an $MnPc/NbS_2$ system are shown in Figure 4b. In an MnPc/graphene system the reaction is endothermic, while for $NbS_2$ and $CoS_2$ the reaction is exothermic. A single linear relation between the final state energies and the transition state energies is shown in Figure 4c, indicating that all transition state energies follow final state behavior: the system which has a lower final state energy shows better catalytic activity in the $H_2$ dissociation reaction. The linear relation can be understood according to the universality principle in heterogeneous catalysis, namely, the Brønsted-Evans-Polanyi (BEP) relation [32]. The $H_2$ entry barrier to the $MnPc/NbS_2$ interface is 0.78 eV, which is smaller than the calculated binding energy (3.2 eV). Hence, metallic transition metal dichalcogenides are good candidates for the quantum nutcracker because the reaction is mediated by the electronic properties of the surface.

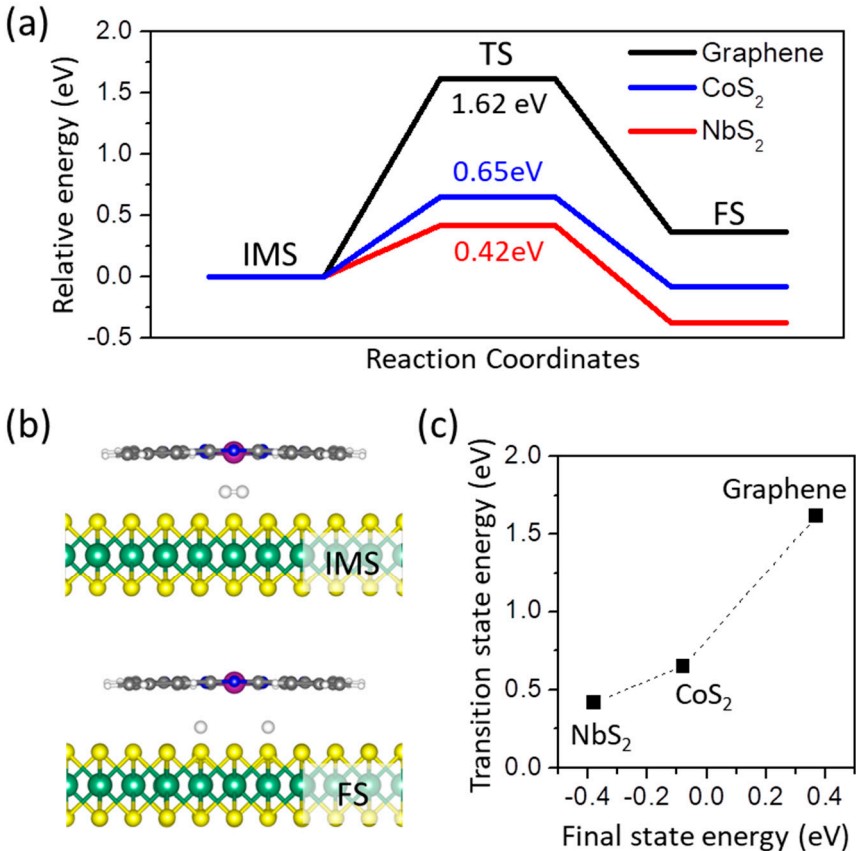

**Figure 4.** (**a**) Reaction pathway of hydrogen dissociation between an MnPc molecule and a monolayer of 2D materials. (**b**) Side views of intermediate and final state of hydrogen dissociation in an MnPc/NbS$_2$ system. (**c**) Transition state energy as a function of the final state energy in a quantum nutcracker.

In summary, in this work we revisited the concept of a quantum nutcracker and investigated the factors that influence its catalytic activity. Using density-functional-theory (DFT) calculations we found that the H$_2$ entry barrier, which is the rate-limiting barrier, is reduced by replacing a transition metal phthalocyanine molecule with a transition metal porphyrin or one of its derivatives, such as OEP and TPP. The quantum nutcracker is still active when a dissociated H atom adsorbs on top of the transition metal ion. Electronic property calculations revealed that the H–H bond splits via interactions with the $d_{xz}$ or $d_{yz}$ orbital of the transition metal ion. We also explored the catalytic activity of a quantum nutcracker based on representative 2D materials. We found that metallic 2D materials, such as transition metal dichalcogenides CoS$_2$ and NbS$_2$, are also good. These results extend the concept of the quantum nutcracker and provide new opportunities for functionalizing catalytically inert materials.

**Author Contributions:** L.T. and Y.-Y.Z. performed the calculations under the guidance of S.D. and S.T.P. The manuscript was written via contributions by all authors. All authors gave approval to the final version of the manuscript.

**Funding:** This work was financially supported by the National Natural Science Foundation of China (nos. 61888102 and 5192201), the National Key Research and Development Projects of China (2016YFA0202300, 2018YFA0305800, and 2019YFA0308500), the Strategic Priority Research Program of the Chinese Academy of Sciences (no. XDB30000000), the CAS Pioneer Hundred Talents Program, K.C. Wong Education Foundation, and the International Partnership Program of the Chinese Academy of Sciences (no. 112111KYSB20160061). Work at Vanderbilt University (L.T. and S.T.P) was supported by the U.S. Department of Energy (grant DE-FG02-09ER46554) and the McMinn Endowment. Computations at Vanderbilt University were carried out at the National Energy Research Scientific Computing Center, a DOE Office of Science User Facility supported by the Office of Science of the U.S. Department of Energy (contract no. DE-AC02-05CH11231).

**Acknowledgments:** All authors would like to thank Wei Guo for critical discussions.

**Conflicts of Interest:** The authors declare no conflict of interest.

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
