# Peer review of "Tuning the Catalytic Activity of a Quantum Nutcracker for Hydrogen Dissociation"

_surfaces, doi:10.3390/surfaces3010004_

Round 1
Reviewer 1 Report
This work describes theoretical findings regarding the dissociation of H2 molecule between metallic surfaces (including transition-metal dichalcogenides CoS2 and NbS2) and metal porphyrin-based molecules.
As a general comment I believe the readers would like to see some discussion on the final catalytic efficiency of the systems which are reported herein in comparison with other catalysts for H2 dissociation. Are there other similar studies on H2 dissociation on some catalytic surfaces? Some statement in the discussion about which system the authors would suggest as more efficient overall among those presented here.
The authors state in lines 119-121 that : "The orbital overlap between the s orbital of the H atoms and the Mn dxz orbital shows that H-H bond breaking happens via the interaction of the hydrogen anti-bonding state with the transition-metal ion." What is the role of the metallic substrate in this system? What would be the result if there was no metallic substrate? Maybe the authors have given answers in previous studies but the presentation could be also complete here as well.
Why was the study on the role of the absorbed hydrogen atom performed on the phthalocyanine system which has been published previously and not on the current materials?
The introduction could be enriched with some examples where the H2 dissociation is useful/applied.
While in lines 51-63 the introduction is concluded with the main finding of the work, the the text between lines ~70-81 is like returning back to introduction.
Text in lines 88-89 should probably be inserted in the figure caption
In line 81 there is a forgotten comment about inserting a reference.
Reviewer 2 Report
The article "Tuning the catalytic activity of quantum nutcracker for hydrogen dissociation" is a theoretical investigation of the hydrogen dissociation at the interface between a surface and a molecule. The concepts and results are interesting although there is no mention to experimental data that could support the theoretical findings.
The authors should introduce references where the feasibility of such process is confirmed experimentally.
the energy necessary to intercalate H2 below the molecule is quite high: i wonder if such can energy is provide the molecule would desorb from the surface. the authors shovel therefore report the adsorption energy of the molecules on the different substrates and comment on these numbers.
More details about the models and computational parameter should be given especially about the CI-NEB) method. Moreover, in the case of MnTPP and OEP how large is the unit cell? are the interactions between neighboring molecules important or negligible?
Minor issues.
the authors refer to the porphyrins and phtalocynine as "organometallic" which is semantically wrong (no direct metal-carbon bonds), the term coordination compounds or organic molecules is more adequate
the caption of fig. 2 on pg 3 rows 88-89 is incorrectly formatted as text
pg 2 row 81 a reference is missing and the text "cite reference here " should be deleted
Round 2
Reviewer 1 Report
Dear Authors and dear Editor,
After the revision by the authors, I would recommend the publication of the manuscript.